# OX40-OX40L Inhibition for the Treatment of Atopic Dermatitis—Focus on Rocatinlimab and Amlitelimab

**DOI:** 10.3390/pharmaceutics14122753

**Published:** 2022-12-08

**Authors:** Ana Maria Lé, Tiago Torres

**Affiliations:** 1Department of Dermatology, Centro Hospitalar Universitário do Porto, 4099-001 Porto, Portugal; 2Instituto de Ciências Biomédicas Abel Salazar, University of Porto, 4050-313 Porto, Portugal

**Keywords:** OX40, OX40-OX40L, rocatinlimab, amlitelimab, atopic dermatitis

## Abstract

Despite the recent emergence of targeted therapeutic options, there are still unmet needs concerning moderate-to-severe atopic dermatitis treatment. This review aims to discuss the OX40-OX40L pathway as a therapeutic target for the treatment of atopic dermatitis. OX40 and OX40L are two checkpoint molecules that bind to potentiate pro-inflammatory T-cell responses that are pivotal to atopic dermatitis pathogenesis. Two OX40-OX40L inhibitors, rocatinlimab and amlitelimab, are being developed for the treatment of atopic dermatitis. Rocatinlimab, an anti-OX40 antibody, was evaluated in phase 2b, a randomized, placebo-controlled clinical trial. At week 16, rocatinlimab groups achieved a greater reduction in the EASI percentage change from the baseline (−48.3% to −61.1%) against the placebo (−15.0%; *p* < 0.001), and clinical response was maintained 20 weeks after the treatment had ceased. Amlitelimab, an anti-OX40L antibody, was studied in a 12-week treatment phase 2a clinical trial, with a significant efficacy response observed within 2 weeks. At week 16, amlitelimab groups reached the EASI mean percentage change from the baseline of −69.9% and −80.1% versus the placebo (−49.4%; *p* = 0.072 and *p* = 0.009). Among the responders, 68% of amlitelimab patients were sustained 24 weeks following the last dose. Both treatments were shown to be safe and well tolerated. Current evidence points to OX40-OX40L inhibitors as future options for atopic dermatitis treatment with potential disease-modifying effects.

## 1. Introduction

Atopic dermatitis (AD) is a chronically relapsing, inflammatory skin disease with a great impact on the patient’s quality of life, especially due to pruritus. With a prevalence of 2–5% in young adults and up to 20% in children, AD is one of the most common skin diseases [1,2].

The currently available treatment armamentarium includes topical therapy (topical corticosteroids, calcineurin inhibitors crisaborole and ruxolitinib), phototherapy, and systemic therapy (conventional immunosuppressants and advanced target therapies). Conventional systemic immunosuppressive therapies may have limited efficacy and harbor long-term toxicity, which makes them not appropriate for continuous use [3,4]. Fortunately, in recent years, several new therapeutic options targeting specific cytokines, cytokine receptors, or intracellular signaling pathways have been developed, showing significant clinical benefits in patients with AD and paving the way for more effective and safer therapies [3,4,5,6]. Dupilumab, a fully human IgG4 monoclonal antibody directed against the IL-4Rα subunit of IL-4 and IL-13 receptors, changed the AD treatment paradigm as it was in 2017 as the first biological drug approved for the treatment of AD [5,6]. Since then, other therapeutic options targeting the IL-13 and janus kinase (JAK) pathway have been approved, such as tralokinumab (IL-13 inhibitor), upadacitinib (JAK1 inhibitor), abrocitinib (JAK1 inhibitor), and baricitinib (JAK1/2 inhibitor) [3,4,7,8], while others, such as lebrikizumab (IL-13 inhibitor) and nemolizumab (IL-31 inhibitor), are in an advanced stage of clinical development.

Nonetheless, some patients are still a therapeutic challenge, whether because they do not respond/lose clinical response or are unable to receive treatment due to tolerability and safety issues. These unmet needs require that new mechanisms of action and new targeted drugs continue to be developed.

This review aims to discuss the OX40-OX40L pathway as a potential therapeutic target for the treatment of AD, focusing on two OX40-OX40L inhibitors: rocatinlimab and amlitelimab.

## 2. AD Pathogenesis and the Role of OX40-OX40L Pathway

A complex interplay between skin barrier dysfunction, skin inflammation, and dysbiosis contributes to AD development and chronicity [9]. The acute phase of AD is characterized by a strong modulation of Th2 and Th22 immune responses, with variable Th17 involvement. With disease chronicity, there is also a marked Th1 activation. Anomalies in filaggrin, intercellular lipids, and tight junctions induce barrier disruption, which leads to the increased permeability of the skin to exogenous stimuli, sebostasis, and increased transepidermal water loss [10,11,12,13]. Epidermis-derived alarmin cytokines, such as thymic stromal lymphopoietin (TSLP), interleukin (IL)-25, and IL-33, respond to environmental insults and mediate intercellular communication between the epidermal keratinocytes and immune cells. These endogenous molecules are rapidly released by keratinocytes in response to tissue damage to trigger defensive immune responses and trigger type 2 immune deviation, with a significantly higher number of T-helper (Th)2 cells expressing IL-4, IL-13, and IL-31. Consecutively, this inflammation downregulates the expression of filaggrin and loricrin in keratinocytes and exacerbates the epidermal barrier dysfunction [3,14]. Furthermore, IL-4 and IL-13 prompt inflammation through the stimulation of IgE production from plasma cells, as well as amplifying IL-31-induced and histamine-induced pruritus [15]. IL-22 is another important disease mediator of the disease, whose expression is increased in the skin of AD patients and mouse models, inducing keratinocyte proliferation and downregulating filaggrin expression [16].

OX40–OX40L interaction (two co-stimulatory immune checkpoint molecules) plays a central role in the pathogenesis of AD. Immune checkpoint molecules have co-stimulatory and co-inhibitory roles in adaptative immune responses. They are primarily classified into two groups: (1) the immunoglobulin superfamily and the (2) the tumor necrosis factor superfamily (TNFSF) and its receptors (TNFRSFs). OX40 (TNFRSF4, CD134) and its ligand OX40L (TNFSF4, CD252) are two of the TNFSF/TNFRSF co-stimulatory immune checkpoint molecules [17].

The co-stimulatory T-cell receptor OX40 is expressed predominantly on effector and regulatory T-cells. Its ligand, OX40L, is expressed on activated antigen-presenting cells, including dendritic cells (DCs), endothelial cells, macrophages, and activated B-cells. OX40–OX40L engagement is key to potentiating the expansion of effector T-cells and the prolongation of their survival by suppressing apoptosis, enhancing T-cell effector functions, such as cytokine production, and generating T helper memory cells. Naïve T-cells are activated by antigen-presenting cells through co-stimulatory molecule interaction, such as CD80/CD86 and CD28. The activated effector Th1 and Th2 T-cell expansion are sustained by OX40-OX40L ligation. Even though resting memory T-cells do not express OX40, upon reactivation, they become effector memory T-cells, and they start expressing it. OX40-OX40L ligation promotes the expansion of these cells [17,18].

Preclinical studies of skin inflammation and asthma models have supported that OX40-OX40L signaling interactions are pivotal to the efficiency of the responses that are regulated by memory Th2 cells [19,20]. TSLP and IL-25 activate DCs to express OX40L. OX40L-positive DCs induce OX40-positive T-cell differentiation, including Th2 cells, which promote IL-4 and IL-13 production from T-cells and signature cytokines of type 2 inflammatory response [14,21]. Il-33 is produced by barrier-disrupted epidermic keratinocytes and stimulates type 2 innate lymphoid cells and dendritic cells to express OX40L. Preclinical evidence suggests that OX40-OX40L signaling also modulates IL-22 production from T-cells (Figure 1) [22].

In patients with AD, the surface expression of OX40 and its ligand on the peripheral blood of mononuclear cells is higher in comparison with healthy adults. Strong correlations were observed between the disease activity scores and Th2-associated markers, TSLPR, and OX40L [18,23].

## 3. OX40-OX40L Inhibition

### 3.1. GBR 830

GBR 830 was the first-in-class, humanized, a monoclonal antibody against OX40 to enter a phase 2a trial investigating the efficacy, safety, and tissue effects in AD patients (NCT02683928) [24,25]. Sixty-two moderate-to-severe AD patients were randomized to 3:1 to 10 mg/kg GBR830 or placebo on day 1 and day 29. On day 71, the proportion of patients achieving a 50% or greater improvement in the Eczema Area and Severity Index (EASI) score was greater with GBR 830 (76.9% [20/26]) versus the placebo (37.5% [3/8]). Biopsy specimens from lesioned skin were obtained before the first dose (baseline) at days 29 and day 71. Significant decreases from the baseline in OX401 T-cell and OX40L1 DC staining in the lesioned skin were found with GBR 830 treatment at days 29 (*p* < 0.05) and 71 (*p* < 0.001). A significant reduction in mRNA cytokines such as IL-31, CCL11, CCL17, and S100 was also demonstrated. GBR 830 was well tolerated, with an equal treatment-emergent adverse events (TEAE) distribution (GBR 830, 63.0% [29/46] and placebo, 63.0% [10/16]). One serious event was reported in the GBR 830 group, but it was deemed unrelated to the study of the drug. The most reported TEAE was a headache, with no clinically meaningful differences between the groups. Myalgias were only reported in the GBR 830 group (6.5% [3/46]) [24,25].

No further studies have been developed with this drug.

### 3.2. Rocatinlimab

Rocatinlimab, formerly known as AMG 451/KHK4083, is a fully human, non-fucosylated, immunoglobulin G1 (IgG1) anti-OX40 monoclonal antibody currently under investigation for the treatment of moderate-to-severe AD. It has been shown to selectively deplete OX40+ activated T-cells and suppress clonal T-cells and is expected to control Th2-driven conditions [26].

A phase 1, single-center, open-label trial evaluated 22 patients with moderate-to-severe AD through a 6-week treatment period with repeated intravenous infusions of 10 mg/kg KHK4083 every 2 weeks (a total of three infusions) and a 16-week follow-up period (NCT03096223) [27]. KHK4083-related infusion reactions of mild or moderate severity were the most reported and included pyrexia (11 patients, 50.0%) and chills (8 patients, 36.4%). Aphthous ulcers (4 patients, 18.2%), blood uric acid increase (3 patients, 13.6%), nasopharyngitis (3 patients, 13.6%), erythema (2 patients, 9.1%), and hordeolum (2 patients, 9.1%) were also described. From a baseline EASI (mean ± SD) of 33.98 ± 9.68, the percent change from the baseline at day 43 was −24.25 ± 27.55% and −74.12 ± 20.53% at day 155 [27].

A phase 2b multicenter, randomized, double-blind, parallel-group, placebo-controlled clinical trial (NCT03703102) was posteriorly developed to further evaluate the efficacy and safety of rocatinlimab in subjects with moderate-to-severe atopic dermatitis with an inadequate response to topical treatments [28]. The 274 subjects were randomly assigned (1:1:1:1:1) rocatinlimab with 150 mg subcutaneous (SC) every 4 weeks (Q4W), 600 mg SC Q4W, 300 mg SC every 2 weeks (Q2W), 600 mg SC Q2W, or a placebo. Rocatinlimab groups received treatment for 36 weeks, followed by an off-drug follow-up period of 20 weeks. The placebo group received a placebo for 18 weeks, followed by an additional 18-week rocatinlimab treatment period (600 mg SC Q2W) and a 20-week off-drug follow-up period. According to the non-peer-reviewed data, at week 16, all rocatinlimab groups achieved a significant reduction in the percentage change from the baseline in EASI score at week 16 (−48.3% to −61.1%) compared to the placebo (−15.0%; all *p* < 0.001). Higher proportions of patients in the rocatinlimab groups achieved EASI75 (44.2%, 40.4%, 53.8%, and 38.9%, respectively, versus the placebo: 10.5%). Among the rocatinlimab patients, 36.5% to 55.8% achieved a 4-point improvement or greater from the baseline in the pruritus NRS score (placebo: 19.3%). On each endpoint, greater improvements were observed at week 16 for rocatinlimab: 300 mg Q2W in comparison to other doses [28]. Efficacy measures continued to improve after week 16 for all rocatinlimab groups, with the highest responses observed for the 300 mg Q2W group (EASI75 = 65.4% and 63.5% at weeks 24 and 36, respectively) [28]. Through week 18, TEAE was reported in 81% of patients in the rocatinlimab group against 72% in the placebo group. The most frequent ones were pyrexia and chills after the first administration of rocatinlimab, nasopharyngitis, and atopic dermatitis (Table 1) [28].

A clinical biomarker sub-study aimed at analyzing transcriptomic and proteomic profiles in skin biopsy specimens and serum samples of the participating AD patients. Skin biopsy specimens were collected from 20/150 Japanese patients at the baseline, week 8, week 16, week 36, and week 52, and the genomic profile showed significant and robust changes from the baseline throughout treatment, approaching that of the non-lesional skin. Quantitative polymerase chain reaction analysis revealed a reduced OX40 mRNA expression and the downregulation of Th2, Th1/Th17, and Th22-related genes after rocatinlimab treatment. The effects of rocatinlimab on gene expression persisted after the discontinuation of treatment at week 36, through week 52. A reduction in Th2/Th22 and pruritus-related molecules were also observed in proteomic analysis using serum samples at week 16 [29].

A post hoc analysis evaluated rocatinlimab’s efficacy concerning head and neck disease. The head and neck EASI score was calculated with the adjustment of 0.1, equating to 10% of the weighting of the head and neck region in the total EASI score. Reported results were consistent with the main analysis as all rocatinlimab doses resulted in greater head and neck EASI score improvements versus the placebo until week 56, 20 weeks after treatment had ceased [30].

A phase 3, 52-week, multicenter, randomized, double-blind, placebo-controlled trial is currently in progress (active, not recruiting) to further evaluate two different doses in four different treatment schemes (NCT05398445-ROCKET-IGNITE) (Table 2).

### 3.3. Amlitelimab

Amlitelimab, also known as KY1005/SAR445229, is another OX40-OX40L pathway inhibitor but with a different mechanism of action. It is a non-depleting IgG4 human anti-OX40L monoclonal antibody that binds OX40L and blocks interactions with OX40. By targeting OX40L, amlitelimab aims to restore immune homeostasis between pro-inflammatory and anti-inflammatory T-cells. In addition to blocking antigen-presenting T-cell activation, amlitelimab also blocks T-cell independent antigens that present the cells’ pro-inflammatory activity via the inhibition of OX40L back signaling, thus blocking both type 2 and Th1/17/22 inflammation [31,32].

A phase 1 trial (NCT03161288) was first conducted with 64 healthy subjects to evaluate its safety, tolerability, and immunogenicity profile. Subjects were enrolled into eight cohorts, and in each of them, they were randomized to the intravenous administration of KY1005 (different doses) or a placebo. Two subjects per cohort started as a sentinel group, and if no safety issues arose within 48 h after dosing, the remaining six subjects received treatment. At 4 and 8 weeks after the initial administration, the subjects received two maintenance doses (50% of the loading dose). One subject in the 12 mg/kg cohort did not receive the second and third KY1005 doses due to a possible mild and self-limited hypersensitivity reaction. All treatment-emergent adverse events were of mild (*n* = 190) or moderate severity (*n* = 16) and were self-resolving without sequalae, a headache being the most reported one. There were no clinically significant changes in any safety laboratory parameters [31].

A phase 2a (NCT03754309) double-blind, randomized, controlled trial was conducted for 16 weeks with 89 moderate-to-severe AD patients who were intolerant or had an inadequate response to topical treatments. They were randomized 1:1:1 to an intravenous amlitelimab low dose (LD, 200 mg loading/100 mg maintenance Q4W, *n* = 29), high dose (HD, 500 mg loading/250 mg maintenance Q4W, *n* = 30), or a placebo (*n* = 29) until week 12. At the end of week 16, there were 59 evaluable patients, and at the end of week 36, there were 50. The main reason for the study discontinuation was a withdrawal of consent (15/29) [32,33].

According to non-peer-reviewed data, at week 16, the mean percentage change from the baseline in EASI was significantly greater in patients receiving amlitelimab LD (−80.1%) and HD (−69.9%) vs. the placebo (−49.4%; *p* = 0.009 and *p* = 0.072, respectively). EASI−75 was reached by 59.3% of patients in the amlitelimab LD group, 51.9% in the amlitelimab HD group, and 25.0% in the placebo group. The onset of response was as early as week 2 for both amlitelimab groups. Additionally, 44% of patients treated with amlitelimab-LD and 37% of patients treated with amlitelimab-HD achieved a score of 0 (clear) or 1 (almost clear) on the validated Investigator’s Global Assessment (vIGA) scale compared with 8% with placebo (*p* < 0.001 both LD and HD). At week 36, 68% of patients who achieved a vIGA score of 0 or 1 at week 16 maintained their response 24 weeks after their last dose. Pruritus NRS ≥ 4-point improvement at week 16 was reached by 57.9% in amlitelimab LD, 62.5% in amlitelimab HD, and 38.1% in the placebo group [33]. Safety follow-up was conducted until week 36, and amlitelimab was globally well tolerated. The overall rate of TEAEs was 35% for amlitelimab-LD, 17% for amlitelimab-HD, and 31% for the placebo. No hypersensitivity or tolerability events were reported, and, despite the incidence of related TEAEs, no clinically relevant pattern of events was observed. (Table 1) [33]

Considering the importance of IL-22 in AD pathogenesis and its hypothesized association with the OX40-OX40L pathway, a specific analysis was performed to evaluate its behavior during treatment. The IL-22 baseline serum levels correlated significantly with the severity of the disease (EASI r = 0.53, *p* < 0.0001; and SCORAD r = 0.36, *p* = 0.001), and no significant differences were observed between the groups. At week 16, a significant decrease in IL-22 serum levels was observed in patients treated with amlitelimab but not in the placebo (*p* = 0.381). An amlitelimab-induced decrease in IL-22 levels was maintained until week 36 in those defined as vIGA 0/1 responders at week 16, 24 weeks after their last dose. On the other hand, there were no significant disparities in the baseline IL-22 between responders and non-responders [32].

A Phase 2b study is now recruiting (STREAM-AD; NCT05131477) to further evaluate the impact of amlitelimab, when given subcutaneously, in patients with moderate-to-severe atopic dermatitis. It is an interventional, randomized, parallel-group, phase 2b, double-blind, 5-arm trial, which aims to assess the effect of amlitelimab in adult patients with moderate-to-severe atopic dermatitis. Additionally, a single group, phase 2, long-term extension study (NCT05492578) will characterize the safety and efficacy of amlitelimab in treated adult participants with moderate-to-severe AD who have previously been enrolled in the study (NCT05131477) (Table 1).

Amlitelimab is also under research for the treatment of moderate-to-severe asthma (TIDE-asthma; NCT05421598).

## 4. Discussion

Atopic dermatitis poses a therapeutic challenge for clinicians, particularly in moderate-to-severe forms of the disease. Systemic corticosteroids and classic immunosuppressants, such as cyclosporine, methotrexate, mycophenolate mofetil, and azathioprine, have their prolonged use limited by safety concerns, variable efficacy response, and the need for frequent laboratory monitoring. Increased knowledge of atopic dermatitis complex pathophysiology has allowed the consideration of more immunological pathways as potential therapeutic targets. The currently available targeted options include biologic therapy, dupilumab, tralokinumab, and oral JAK inhibitors. Dupilumab, an IL-4 and IL-13 pathway inhibitor, was the first available biologic agent for the treatment of atopic dermatitis, and it is currently approved by FDA for adults and children aged 6 months and older with moderate to severe diseases that are not adequately controlled with topical therapies. Tralokinumab is an anti-IL13 antibody recently approved for adult patients. JAK inhibitors, such as baricitinib, upadacitinib and abrocitinib, are efficient new oral small molecules but with a less specific mechanism of action, raising more safety concerns, especially in high-risk patients [34,35,36]. OX40-OX40L interaction is an essential part of the immune cascade that allows T-cell functioning and Th1, Th2, and Th22-mediated pathways, which have all been implicated in AD.

GBR 830 was the first-in-class monoclonal antibody against OX40 to present good efficacy and safety results, but its investigation has not been pursued. Rocatinlimab, an anti-OX40 monoclonal antibody, recently completed a phase 2b trial in which four different doses were evaluated, all of them with good efficacy results. The patients kept improving until week 36 of treatment. A post hoc head and neck analysis revealed that the therapeutic effect seemed to persist for 5 months after the treatment had ceased, which was supported by skin transcriptomic analysis. This durability of the response with rocatinlimab potentially reflects how this mechanism of action may induce disease modification [30]. Rocatinlimab 300 mg Q2W was the dosage scheme that obtained greater improvements in all efficacy endpoints. However, administrations every 4 weeks may be more convenient and improve compliance. A phase 3 trial is currently in progress, aiming to evaluate two different doses every 2 weeks for 24 weeks, followed by every 4 weeks for 28 weeks. It will provide further information on which dose and periodicity are more adequate and whether extended dosing is feasible.

Amlitelimab is another OX40-OX40L pathway inhibitor that binds OX40L and blocks the interaction with OX40. It presented a rapid and marked clinical improvement in patients with moderate-to-severe AD, with a good safety profile. These efficacy results were maintained for 6 months after the treatment ceased, which also could suggest a disease-modifying effect. This could indicate a long and sustained response following the last dose, opening up the opportunity for extended dosing. The sustained reduction in IL-22 serum levels in amlitelimab-treated patients strongly indicates that amlitelimab effectively targets immune dysregulation in AD. It supports the hypothesis that targeting OX40L on antigen-presenting cells modulates not only type 2 response but also other T-cell pathways, including Th22 [32].

Both rocatinlimab and amlitelimab presented no major safety issues. Pyrexia and chills, after the first administration of rocatinlimab, were frequent, but they were not reported in subsequent administrations. No hypersensitivity or tolerability events were reported with amlitelimab. However, long-term studies will be essential to determine the potential risks. Animal OX40 deficient models seem to have impaired interferon-γ production and Th1 differentiation. On the other hand, in the viral infection model, virus-specific antibody production and the virus-specific cytotoxic T-cell response were not affected. Ongoing long-term extension and phase 3 trials will clarify whether the increased risk of infections will be a concern [17].

The evidence thus far points to OX40-OX40L inhibitors as a future efficient and safe option for the treatment of AD. Their potential disease-modifying effect could be ground-breaking and life-changing for our patients. Peer-reviewed information and further investigation with large phase 3 trials will be fundamental to understanding this class of drugs’ positioning in the atopic dermatitis therapeutic armamentarium.

## Figures and Tables

**Figure 1 pharmaceutics-14-02753-f001:**
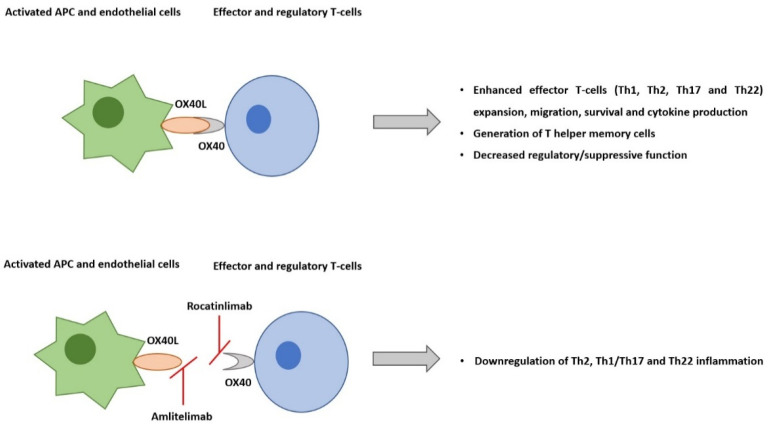
The co-stimulatory T-cell receptor OX40 is expressed predominantly on effector and regulatory T-cells. Its ligand, OX40L, is expressed on activated antigen-presenting cells and endothelial cells. Their interaction plays a central role in pro-inflammatory T-cell responses, including in atopic dermatitis. Rocatinlimab is a fully human anti-OX40 monoclonal antibody, which selectively depletes sOX40+ activated T-cells. On the other hand, amlitelimab, is a non-depleting IgG4 human anti-OX40L monoclonal antibody that binds OX40L and blocks interaction with OX40. Besides blocking antigens, the presenting T-cell activation, amlitelimab also blocks T-cell independent APC pro-inflammatory activity via the inhibition of OX40L back signaling.

**Table 1 pharmaceutics-14-02753-t001:** Main characteristics and results of clinical trials concerning rocatinlimab and amlitelimab.

Trial	Phase	Study Design	Primary Endpoints	Main Results
Rocatinlimab NCT03096223	1	Single-center, open-label 22 subjects with moderate-to-severe AD Rocatinlimab 10 mg/kg intravenous Q2W 6 weeks of treatment + 16 weeks of follow-up	Incidence of treatment-emergent adverse events up to week 22	Treatment-emergent adverse events: rocatinlimab-related infusion reactions (mild or moderate severity): pyrexia (11 patients, 50%) and chills (8 patients, 36.4%) EASI change from baseline, % (mean ± SD): - 24.25 ± 27.55% at day 43 - 74.12 ± 20.53% at day 155
Rocatinlimab NCT03703102	2b	Multi-center, double-blind, placebo-controlled 274 subjects with moderate-to-severe AD Randomized 1:1:1:1:1 to: rocatinlimab 150 mg SC Q4W rocatinlimab 600 mg SC Q4W rocatinlimab 300 mg SC Q2W rocatinlimab 600 mg SC Q2W placebo 18 weeks of treatment + 20 weeks of follow-up	% EASI change from baseline at week 16	% EASI change from baseline at week 16 (-48.3% to -61.1%) vs. placebo (-15.0%; all *p* < 0.001). ≥4-point improvement from baseline in pruritus NRS score (36.5% to 55.8%) vs. placebo (19.3%) A *post hoc* analysis reported EASI score improvements up to 20 weeks after treatment has ceased At week 18, most treatment-emergent adverse events were pyrexia and chills after the first administration of rocatinlimab, nasopharyngitis, and atopic dermatitis
Amlitelimab NCT03161288	1	Single-center, open-label, randomized, parallel group 64 healthy subjects Subjects were enrolled into 8 cohorts and, in each cohort, they were randomized to amlitelimab or placebo (6:2).	All treatment-related adverse events; changes in vital signs, laboratory safety data, anti-viral antibody levels and viral DNA, acute cytokines and in electrocardiograms.	All treatment emergent adverse events were of mild or moderate severity, without sequalae (++ headache). There were no clinically significant changes in any safety laboratory parameters or other safety concerns.
Amlitelimab NCT03754309	2a	Multi-center, parallel group, double-blind, randomized, placebo controlled 89 moderate-to-severe AD patients Randomized 1:1:1 to: - amlitelimab 200 mg loading dose + 100 mg Q4W - amlitelimab 500 mg loading dose + 250 mg Q4W - placebo 12 weeks of treatment + 24 weeks of follow-up	% EASI change from baseline to day 113 Incidence of treatment-emergent adverse events	Mean percentage change from baseline in EASI ate week 16: amlitelimab low-dose (−80.1%) and high-dose (−69.9%) vs. placebo (−49.4%; *p* = 0.009 and *p* = 0.072, respectively). % EASI-75: 59.3% in amlitelimab low-dose group, 51.9% in amlitelimab high-dose group and 25.0% in placebo group. Pruritus NRS ≥ 4-point improvement at week 16: 57.9% in amlitelimab low-dose, 62.5% in amlitelimab high-dose, and 38.1% in placebo group. No hypersensitivity or tolerability events were reported.

AD—atopic dermatitis; EASI—Eczema Area and Severity Index; NRS—numerical rating scale; Q2W—every 2 weeks; Q4W—every 4 weeks; SC—subcutaneous administration.

**Table 2 pharmaceutics-14-02753-t002:** Ongoing clinical trials of OX40-OX40L inhibitors for atopic dermatitis.

Clinical Trial	Drug	Phase	Status
NCT05398445-ROCKET-IGNITE	**Rocatinlimab**	Phase 3	Active, not recruiting
NCT05131477-STREAM-AD	**Amlitelimab**	Phase 2b	Recruiting
NCT05492578–Long-term extension	**Amlitelimab**	Phase 2	Recruiting

## Data Availability

Not applicable.

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
