# Peer review of "OX40-OX40L Inhibition for the Treatment of Atopic Dermatitis—Focus on Rocatinlimab and Amlitelimab"

_pharmaceutics, 2022, doi:10.3390/pharmaceutics14122753_

Round 1
Reviewer 1 Report
Line 40: change " Dupilumab, an IL-4 and IL-13 inhibitor" with "Dupilumab is a fully human IgG4 monoclonal antibody directed against the IL-4Rα subunit of IL-4 and IL-13 receptors"
Line 56: "The acute phase of AD is characterized 56 by a strong modulation of Th2 and Th22 immune responses" but also Th17 and Th1 Czarnowicki T, He H, Krueger JG, Guttman-Yassky E. Atopic dermatitis endotypes and implications for targeted therapeutics. J Allergy Clin Immunol. 2019 Jan;143(1):1-11
Line 188: "It is a non-depleting IgG4" could you provide same information also for other treatments?
Author Response
Responses to Referee´s comments about manuscript pharmaceutics-1988087
Title: OX40-OX40L inhibition for the treatment of atopic dermatitis – focus on rocatinlimab and amlitelimab
Corresponding authors: Tiago Torres, MD, PhD
The authors are grateful for the opportunity to amend some points of the manuscript according to the appreciated referees’ comments, which greatly contributed to improve the manuscript’s quality.
In the following file the authors aim to provide itemized answers to each referee´s comments.

Reviewer 2 Report
In the present review, the authors aim to discuss OX40-OX40L pathway as a therapeutic target for the treatment of atopic dermatitis. OX40 and OX40L are two checkpoint molecules that regulate the pro-inflammatory T-cell responses that are key to atopic dermatitis pathogenesis. In particular, the efficacy and safety of the two OX40-OX40L inhibitors, rocatinlimab and amlitelimab, under development for the treatment of atopic dermatitis, are discussed.
The review is interesting, however, some improvements would certainly increase the interest of the readers of Pharmaceutics.
-The authors should improve the discussion about the current treatment strategies employed for atopic dermatitis.
-A more focused discussion about the mechanism of action of the OX40-OX40L pathway should be included
-The authors could also mention other potential therapeutic interventions, in particular checkpoint inhibitors, currently under evaluation in atopic dermatitis
-A discussion about the limitations for using OX40-OX40L inhibitors would also benefit the review
Author Response

(The authors gave the same response as above.)

Round 2
Reviewer 2 Report
Issues have been addressed